# Deep Multimodal Multilinear Fusion with High-order Polynomial Pooling

**Ming Hou**[1,*], **Jiajia Tang**[2,1,*], **Jianhai Zhang**[2], **Wanzeng Kong**[2], **Qibin Zhao**[1,†]
[1] Tensor Learning Unit, Center for Advanced Intelligence Project, RIKEN, Japan
[2] College of Computer Science, Hangzhou Dianzi University, China
ming.hou@riken.jp, hdutangjiajia@163.com
jhzhang@hdu.edu.cn, kongwanzeng@hdu.edu.cn, qibin.zhao@riken.jp

## Abstract

Tensor-based multimodal fusion techniques have exhibited great predictive performance. However, one limitation is that existing approaches only consider bilinear or trilinear pooling, which fails to unleash the complete expressive power of multilinear fusion with restricted orders of interactions. More importantly, simply fusing features all at once ignores the complex local intercorrelations, leading to the deterioration of prediction. In this work, we first propose a polynomial tensor pooling (PTP) block for integrating multimodal features by considering high-order moments, followed by a tensorized fully connected layer. Treating PTP as a building block, we further establish a hierarchical polynomial fusion network (HPFN) to recursively transmit local correlations into global ones. By stacking multiple PTPs, the expressivity capacity of HPFN enjoys an exponential growth w.r.t. the number of layers, which is shown by the equivalence to a very deep convolutional arithmetic circuits. Various experiments demonstrate that it can achieve the state-of-the-art performance.

## 1 Introduction

Multimodal representation learning has been a very actively growing research field in artificial intelligence and human communication analysis. Its applications have proliferated across human multimodal tasks such as emotion recognition [2], personality traits recognition [22] and sentiment analysis [18]. The multimodal signals collected from diverse modalities (spoken language, visual and acoustic signals) exhibit properties of consistency and complementarity [28]. Extensive studies are dedicated to modelling the multiple modalities and their complex interactions [28, 14, 15, 13]. These interactions are hard to model due to the factors like non-trivial multimodal alignment and unreliable or contradictory information among modalities. It yet remains a major challenge on improving the generalization ability of the model by exploring heterogeneous properties of the multimodal data.

The very key step of multimodal modelling is referred as multimodal fusion, with the aim at integrating features of multiple modalities for yielding more robust predictions. Typically, the multimodal feature fusion can be categorized as early, late and hybrid fusion [1]. Among those, early fusion utilizes the concatenated signals from different sources as the model input [7]. Late fusion, on the other hand, attempts to model each modality separately and thus merge them at the decision level, either by voting or averaging [19, 25]. In hybrid fusion, the output depends on both the predictions of unimodal and the early fusion. Despite being simple, aforementioned conventional fusion techniques are all restricted to the concatenation or averaging of, or more generally, linear combination of multimodal features. And the linear modelling may not be sufficient to capture the complicated intercorrelations.

---

[*]The authors contribute equally
[†]The corresponding author

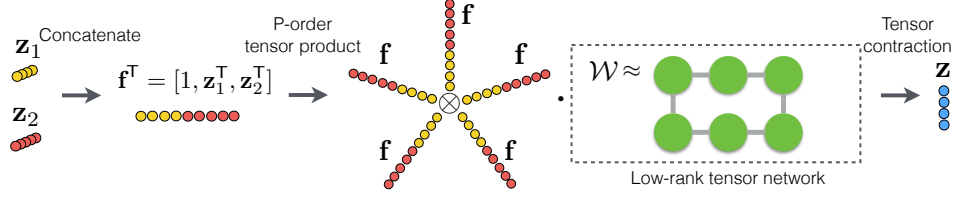

Figure 1: The scheme of 5-order polynomial tensor pooling (PTP) block for fusing $\mathbf{z}_1$ and $\mathbf{z}_2$.

By leveraging tensor product representations, recent fusion models [16, 27] are geared towards modelling bilinear/trilinear cross-modal interactions and boost the performance significantly. Nevertheless, such representations suffer from an exponential growth in feature dimensions, with regard to both the unimodal's dimensionality and the number of modalities, producing a tremendous amount of parameters. To tackle this, the work [17] efficiently reduces fusion parameters by learning low-rank tensor factors, while preserving the capacity of expressing the trimodal (trilinear) interactions.

However, their model fails to unleash the full representation power of multilinear feature intercorrelations by restricting the order of interactions. In other words, the interaction is linear w.r.t. each modality, e.g., only up to trilinear interactions for three modalities. More importantly, their framework focuses on simply fusing multimodal features all at once, totally ignoring the local dynamics of interactions that are crucial to the final prediction. The evolving temporal-modality correlations thus cannot be grasped, which may lead to a deteriorated prediction, especially when long time series are involved.

In this work, we start by proposing a polynomial tensor pooling (PTP) block that can fuse locally mixed temporal-modality features. PTP allows for the higher order moments to capture complex nonlinear multimodal correlations. Building upon the basic PTP block, we further establish a hierarchical architecture that recursively integrates and transmits the local temporal-modality correlations into global ones. This way, fusing multimodal time series data becomes feasible. We refer to the proposed framework as hierarchical polynomial fusion network (HPFN). Using our HPFN brings dual benefits: 1) the local interactions can be grasped at a much finer granularity, and the dominant local correlations can be efficiently transmitted to the global scale. 2) an exponential growth of the expressivity capacity can be achieved by stacking PTPs into multiple layers, which is shown by a connection of HPFN to a very deep convolutional arithmetic circuits. We verify the superior performance of HPFN on two multimodal tasks.

## 2 Preliminaries

We refer multiway arrays of real numbers as *tensors* [12]. We denote a $P$-order tensor $\mathcal{W} \in \mathbb{R}^{I_1 \times \cdots \times I_P}$ with $P$ *modes*. The $(i_1, ..., i_P)$-th entry of $\mathcal{W}$ is denoted as $\mathcal{W}_{i_1,...,i_P}$ with $i_p \in [I_p]$ for all $p \in [P]$, in which the expression $[P]$ represents the set $\{1, 2, ..., P\}$. The *tensor product* denoted as $\otimes$ is a fundamental operator in tensor analysis. Given two tensors $\mathcal{A} \in \mathbb{R}^{I_1,...,I_P}$ and $\mathcal{B} \in \mathbb{R}^{I_{P+1},...,I_{P+Q}}$, the tensor product produces a $(P + Q)$-order tensor $\mathcal{A} \otimes \mathcal{B} \in \mathbb{R}^{I_1,...,I_{P+Q}}$ as

$$\mathcal{A} \otimes \mathcal{B} = \mathcal{A}_{I_1,...,I_P} \cdot \mathcal{B}_{I_{P+1},...,I_{P+Q}}. \tag{1}$$

The tensor product reduces to the standard outer product for the vector inputs. A tensor product of $P$ vectors $\mathbf{v}^{(p)} \in \mathbb{R}^{I_p}$ for $p \in [P]$ yields a rank-1 tensor $\mathcal{A} = \mathbf{w}^{(1)} \otimes \cdots \otimes \mathbf{w}^{(P)}$. The CANDECOMP/PARAFAC (CP) decomposition [3] of $\mathcal{W}$ can be written as a sum of rank-1 tensors as $\mathcal{W} = \sum_{r=1}^{R} \mathbf{w}_r^{(1)} \otimes \cdots \otimes \mathbf{w}_r^{(P)}$, where $R$ is defined as *tensor rank*. *Tensor networks* (TNs) [4] generalize tensor decompositions by factorizing a higher order tensor into a set of sparsely interconnected lower order tensors. TN representation greatly diminishes the effect of curse of dimensionality related to high-order dense tensors. TNs include a number of special cases such as CP, Tucker [26], tensor train (TT) [21] and tensor ring (TR) [32] formats.

## 3 Methodology

We start this section by presenting a product pooling strategy named polynomial tensor pooling (PTP) that serves as a basic building block for our hierarchical polynomial fusion framework (HPFN).

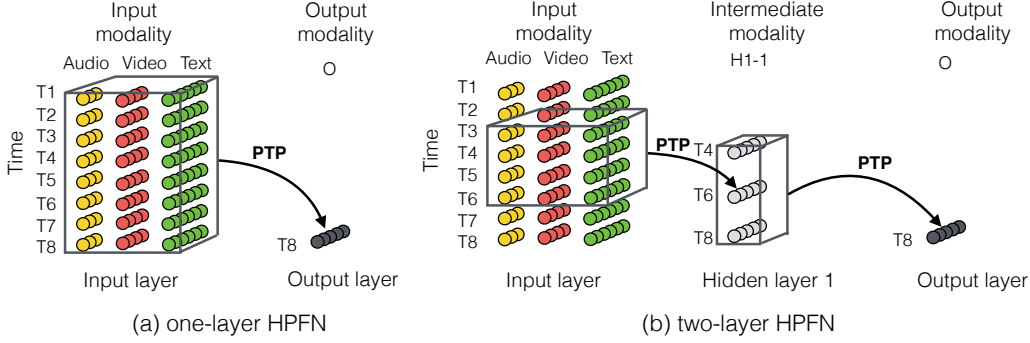

|  |  |
|---|---|
| (a) one-layer HPFN | (b) two-layer HPFN |

Figure 2: (a) An illustrative example of a fusion network with a single PTP block, whose receptive 'window' size is $[8 \times 3]$. (b) An example of two-layer HPFN. For the input layer, the overlapped 'window' has size $[4 \times 3]$ with stride step size 2 along time dimension. For the hidden layer, the 'window' with size $[3 \times 1]$ covers all the intermediate features from the previous layer. H1-1 stands for the '1st' column index of feature nodes in the '1st' hidden layer.

The motivations for PTP are twofold: 1) it explicitly model high-order nonlinear intra-modal and cross-modal interactions; 2) for multimodal time series, it can directly model local interactions within a scanning receptive 'window' across both temporal and modality dimensions.

### 3.1 High-order polynomial tensor pooling (PTP)

The objective of a PTP block is to efficiently merge a collection of features $\{\mathbf{z}_m\}_{m=1}^M$ into a joint compact representation $\mathbf{z}$ by exploiting the explicit interactions of high-order moments. Figure 1 depicts the flowchart of operations in a PTP block. More specifically, a set of $M$ feature vectors $\{\mathbf{z}_m\}_{m=1}^M$ are first concatenated together into a long feature vector $\mathbf{f}$:

$$\mathbf{f}^\mathsf{T} = [1, \mathbf{z}_1^\mathsf{T}, \mathbf{z}_2^\mathsf{T}, \cdots, \mathbf{z}_M^\mathsf{T}]. \tag{2}$$

Then, a degree of $P$ *polynomial feature tensor* $\mathcal{F}$ is formulated using a $P$-order tensor product of the concatenated feature vector $\mathbf{f}$ as

$$\mathcal{F} = \underbrace{\mathbf{f} \otimes \mathbf{f} \otimes \cdots \otimes \mathbf{f}}_{P\text{-order}}, \tag{3}$$

where $\otimes$ is the tensor product operator. Notice $\mathcal{F}$ is capable of representing all possible polynomial expansions up to order $P$ due to the incorporation of the constant term '1' in (2). The effect of $P$ polynomial interaction between features is transformed by a *pooling weight tensor* $\mathcal{W} = [\mathcal{W}^1, ..., \mathcal{W}^h, ..., \mathcal{W}^H]$ as:

$$z_h = \sum_{i_1, i_2, \cdots, i_P} \mathcal{W}^h_{i_1 i_2 \cdots i_P} \cdot \mathcal{F}_{i_1 i_2 \cdots i_P}, \tag{4}$$

where $z_h$ indicates the $h$-th element of the $H$-dimensional fused vector $\mathbf{z}$, while $i_p$ indices the high-order terms in $p$-th mode. Unfortunately, the number of parameters of $\mathcal{W}^h$ in (4) grows exponentially with the polynomial order $P$. To tackle this issue, we adopt the low-rank TNs to efficiently approximate the $\mathcal{W}^h$. Suppose $\mathcal{W}^h$ admits a rank-$R$ CP format, then (4) becomes

$$z_h = \sum_{i_1, i_2, \cdots, i_P} \left[ \left( \sum_{r=1}^R a_r^h \prod_{p=1}^P \mathbf{w}_{r;i_p}^{h(p)} \right) \left( \prod_{p=1}^P \mathbf{f}_{i_p} \right) \right] = \sum_{r=1}^R a_r^h \prod_{p=1}^P \sum_{i_p}^I \mathbf{w}_{r;i_p}^{h(p)} \mathbf{f}_{i_p}. \tag{5}$$

Since the explicitly constructed feature tensor is super symmetric, it then makes sense to assume $\mathbf{w}_r^h = \mathbf{w}_r^{h(p)}$ for all $p \in [P]$. Hence, the $\{\{a_r^h, \mathbf{w}_r^h\}_{r=1}^R\}_{h=1}^H$ are the collection of fusion parameters to estimate. If $\mathcal{W}^h$ admits a TR format, then the following formula can be derived from (4) as

$$z_h = \sum_{i_1, i_2, \cdots, i_P} \left[ \left( \sum_{r_1, r_2, \cdots, r_P} \prod_{p=1}^P \mathcal{G}_{r_p; i_p; r_{p+1}}^{h(p)} \right) \left( \prod_{p=1}^P \mathbf{f}_{i_p} \right) \right]$$

$$= \sum_{r_1, r_2, \cdots, r_P} \prod_{p=1}^P \sum_{i_p}^I \mathcal{G}_{r_p; i_p; r_{p+1}}^{h(p)} \mathbf{f}_{i_p} = \sum_{r_1, r_2, \cdots, r_P} \prod_{p=1}^P \tilde{\mathbf{G}}_{r_p; r_{p+1}}^{h(p)} = \text{Trace} \left( \prod_{p=1}^P \tilde{\mathbf{G}}^{h(p)} \right), \tag{6}$$

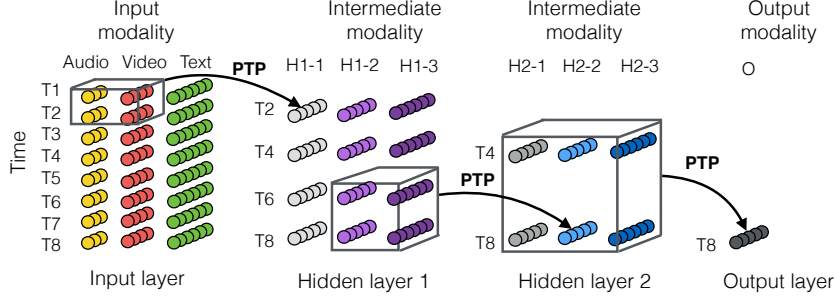

Figure 3: An example of a three-layer HPFN.

where the 3rd-order core tensors $\{\{\mathcal{G}^{h(p)}\}_{p=1}^{P}\}_{h=1}^{H}$ are the fusion parameters. $\{r_p\}_{p=1}^{P}$ are defined as TR-ranks with $r_{P+1} = r_1$. It is also reasonable to assume a shared $\mathcal{G}^h = \mathcal{G}^{h(p)}$ for all $p \in [P]$. In this manner, the fusion computations can be efficiently carried out along each dimension implicitly, thus avoiding the curse of dimensionality on both feature and weight tensors.

## 3.2 Hierarchical polynomial fusion network (HPFN)

Having introduced our basic pooling block, we move on to present the general framework for fusing multimodal data. Generally, if we rearrange multimodal time series as a '2D feature map', the patterns of correlations may manifest themselves in a receptive 'window' covering a local mixture of temporal-modality features across both dimensions. Then, interactions can be gauged by associating a single PTP block to that local 'window'. Using a hierarchical architecture, the local temporal-modality patterns of correlations can be recursively integrated via stacking PTPs in multiple layers. At the end, significant correlations are identified and transmitted to the global scale.

Figure 2 (a) shows a simple one-layer fusion network, with a single PTP operating on one receptive 'window' that covers features across all $8$ time steps and $3$ modalities. This way, PTP makes it feasible to capture the high-order nonlinear interactions among the total $24$ mixed features within the 'window'. We observe a PTP naturally characterizes local correlations if it is linked to a small receptive 'window'. And several PTP blocks can be placed on the local 'windows' of mixed features at distinct locations in a '2D feature map'. It is then straightforward to distribute the fusion process into a number of layers by attaching PTP blocks to small 'windows' at each layer. In fact, the fused node in higher layer corresponds to a larger effect receptive 'window' of features at the lower layer. As a result, more expressive local and global correlations can be efficiently modelled with a great flexibility. The proposed framework is termed as hierarchical polynomial fusion network (HPFN).

Figure 3 displays an instance of three-layer HPFN. At the first hidden layer, each PTP attempts to model local interactions in a 'window' of 2 time steps and 2 modalities. For instance, the audio and video features spanning time T1 and T2 are merged into the resulting hidden node H1-1 at time T2; likewise, the hidden node H1-3 at time T2 is outputted by fusing audio and text features of T1 and T2. The second hidden layer is fed with intermediate features of the previous layer. At the output layer, the final feature is obtained by employing PTP on the intermediate features of 3 modalities in second hidden layer for the time T4 and T8.

Due to the flexibility of our HPFN, various choices for the architecture design are possible. In principle, adding more intermediate layers leads to more complicated and higher-order interactions within a much larger effective receptive 'window'. More complex interactions can also be modelled by allowing the 'windows' to be overlapped. Figure 2 (b) demonstrates an architecture of two-layer HPFN where the fusing 'windows' of $[4 \times 3]$ are overlapped at a stride size of $2$ along the time dimension. More variations can be realized by making an analogy of our PTP to a convolution filter. Just like CNN, a PTP operator can viewed as a 'fusion filter'. In this way, our HPFN may also borrow some similar benefits from the architecture of regular CNN. More precisely, at each layer the PTP 'fusion filter' could be shared when the scanning 'window' slides along the time dimension, so as to catch the important patterns of correlations repeated in time series. Furthermore, associating several PTP 'fusion filters' with one 'window' at the same time is able to capture multiple patterns of correlations existing in that 'window'.

The empirical success of densely connected networks (DenseNets) [11] serves another inspiration to extend HPFN architecture. The incorporation of dense connectivity enhances the expressive capacity

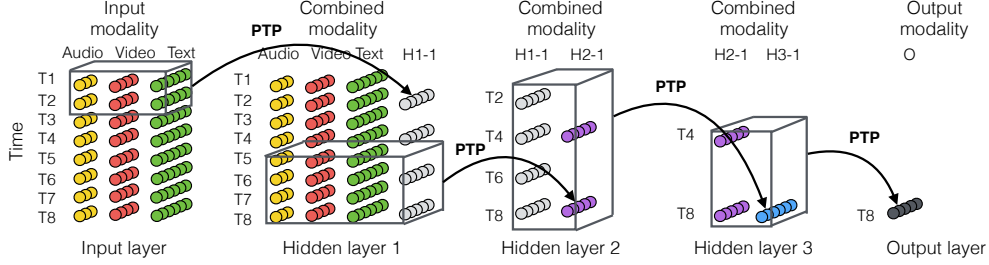

Figure 4: An example of densely connected fourth-layer HPFN with growth rate $k = 1$.

of the fusion model. Adding dense inter-connections could be beneficial in dealing with sequential signals. Specifically, dense connectivity is realizable via the direct inclusion of the features from previous layers into the current layer. The number of previous layers $k \in \mathbb{N}$ involved in connections is defined as the growth rate. Figure 4 depicts an instance of dense HPFN with growth rate $k = 1$.

### 3.3 Connections to convolutional arithmetic circuits

It is interesting to observe that equation (5) suggests PTP actually conducts a combined operations of convolution, pooling and linear transformation. This is quite analogous to convolutional arithmetic circuits (ConvACs) [5] which can be seen as special variants of CNNs. Rather than the rectifier activation and average/max pooling, ConACs are equipped with linear activation and product pooling layers. The authors of [5] analyze the expressivity capacity of the deep ConACs by deriving their equivalence with the hierarchical tucker decomposition (HTD) [9]. It has been proved that deep ConvACs enjoy a greater expressive power than the regular rectifier based CNNs [5]. In fact, a single PTP block corresponds to a shallow ConvAC if the CP format is utilized, and further corresponds to a deep ConAC if the HTD is adopted for the pooling weight tensor. The major difference between ConAC and PTP is that, the product pooling of the standard ConAC is conducted over the locations of features, whereas the product pooling of PTP is over the polynomial orders of concatenated features. Stacking PTP blocks into multiple layers is essentially equivalent to employing multiple HTDs in a recursive manner, resulting in a correspondence of our HPFN to a even deeper ConAC. As a consequence, more flexible higher-order local and global intercorrelations can be explicitly and implicitly captured by HPFN, whose great expressive power can be implied by the connection of HPFN to a very deep ConAC.

### 3.4 Model complexity

This section compares the model complexity of HPFN with two other tensor based models: TFN [27] and LMF [17]. As for PTP, exploiting the symmetry property of the feature tensor, the number of parameters in weight tensor is independent of order $P$, and linearly scales with the concatenated mixed features in 'windows'. For $L$-layer HPFN, the amount of parameters is linearly related to the total number of PTP 'windows' $\sum_{l=1}^{L} N_l$, where $N_l$ is the number of 'windows' at layer $l \in [L]$. In practice, $N_l$ is usually small and decreasing along layers, e.g. $N_1 > N_2 > \cdots > N_L$. Adopting the sharing strategy along the time dimension makes $N_l$ even smaller. In principle, as referred in Table 1, the parameter of HPFN is larger than or comparable to LMF, but significantly less than that of TFN.

Table 1: Model complexity comparisons of TFN, LMF and our HPFN. $I_y$ is the output feature length. $M$ is the number of modalities. $R$ is the tensor rank. For PTP and HPFN, [ $T, S$ ] is the local 'window' size with $S \leq M$ . $I_{t,m}$ is the dimension of features from modality $m$ at time $t$.

| Model | TFN [non-temporal] | LMF [non-temporal] | PTP [temporal] | HPFN (L layers) [temporal] |
|---|---|---|---|---|
| Param. | $\mathcal{O}(I_y \prod_{m=1}^{M} I_m)$ | $\mathcal{O}(I_y R(\sum_{m=1}^{M} I_m))$ | $\mathcal{O}(I_y R(\sum_{t=1}^{T} \sum_{m=1}^{S} I_{t,m}))$ | $\mathcal{O}(I_y R(\sum_{l=1}^{L} N_l)(\sum_{t=1}^{T} \sum_{m=1}^{S} I_{t,m}))$ |

## 4 Related work

There exist two major lines of multimodal fusion research: **non-temporal models** summarize the observations of each unimodal by averaging the features along the temporal dimension. These models have found their utility in the early work of multimodal sentiment analysis [18, 31]. Recently, tensor fusion network (TFN) [27] exploits tensor product to model non-temporal unimodal, bimodal and

Table 2: Specifications of network architecture for non-temporal version of HPFN. [-] indicates the configuration of a specific layer. $\texttt{PTP}_m^k$ denotes the '$m$'th fused feature node in the layer '$k$'.

| Model | Description of Layer-wise Configuration |
|---|---|
| HPFN | $[\texttt{PTP}_1^o(\texttt{a},\texttt{v},\texttt{1})]$ |
| HPFN-L2 | $[\texttt{PTP}_1^{h1}(\texttt{a},\texttt{v}), \texttt{PTP}_2^{h1}(\texttt{v},\texttt{1}), \texttt{PTP}_3^{h1}(\texttt{a},\texttt{1})] - [\texttt{PTP}_1^o(\texttt{PTP}_1^{h1}, \texttt{PTP}_2^{h1}, \texttt{PTP}_3^{h1})]$ |
| HPFN-L2-S1 | $[\texttt{PTP}_1^{h1}(\texttt{a},\texttt{v},\texttt{1})] - [\texttt{PTP}_1^o(\texttt{PTP}_1^{h1}, \texttt{a}, \texttt{v}, \texttt{1})]$ |
| HPFN-L2-S2 | $[\texttt{PTP}_1^{h1}(\texttt{a},\texttt{v}), \texttt{PTP}_2^{h1}(\texttt{v},\texttt{1}), \texttt{PTP}_3^{h1}(\texttt{a},\texttt{1})] - [\texttt{PTP}_1^o(\texttt{PTP}_1^{h1}, \texttt{PTP}_2^{h1}, \texttt{PTP}_3^{h1}, \texttt{a}, \texttt{v}, \texttt{1})]$ |
| HPFN-L3 | $[\texttt{PTP}_1^{h1}(\texttt{a},\texttt{v}), \texttt{PTP}_2^{h1}(\texttt{v},\texttt{1}), \texttt{PTP}_3^{h1}(\texttt{a},\texttt{1})] -$ $[\texttt{PTP}_1^{h2}(\texttt{PTP}_1^{h1}, \texttt{PTP}_2^{h1}), \texttt{PTP}_2^{h2}(\texttt{PTP}_1^{h1}, \texttt{PTP}_3^{h1}), \texttt{PTP}_3^{h2}(\texttt{PTP}_2^{h1}, \texttt{PTP}_3^{h1})] - [\texttt{PTP}_1^o(\texttt{PTP}_1^{h2}, \texttt{PTP}_2^{h2}, \texttt{PTP}_3^{h2})]$ |
| HPFN-L4 | $[\texttt{PTP}_1^{h1}(\texttt{a},\texttt{v}), \texttt{PTP}_2^{h1}(\texttt{v},\texttt{1}), \texttt{PTP}_3^{h1}(\texttt{a},\texttt{1})] -$ $[\texttt{PTP}_1^{h2}(\texttt{PTP}_1^{h1}, \texttt{PTP}_2^{h1}), \texttt{PTP}_2^{h2}(\texttt{PTP}_1^{h1}, \texttt{PTP}_3^{h1}), \texttt{PTP}_3^{h2}(\texttt{PTP}_2^{h1}, \texttt{PTP}_3^{h1})] -$ $[\texttt{PTP}_1^{h3}(\texttt{PTP}_1^{h2}, \texttt{PTP}_2^{h2}), \texttt{PTP}_2^{h3}(\texttt{PTP}_1^{h2}, \texttt{PTP}_3^{h2}), \texttt{PTP}_3^{h3}(\texttt{PTP}_2^{h2}, \texttt{PTP}_3^{h2})] - [\texttt{PTP}_1^o(\texttt{PTP}_1^{h3}, \texttt{PTP}_2^{h3}, \texttt{PTP}_3^{h3})]$ |

trimodal interactions between modalities. To handle the curse of dimensionality issue, low-rank multimodal fusion network (LMF) [17] further enhances the scalability of non-temporal fusion with modality-specific low-rank factors. All those approaches, with the averaged statistics of features, attempt to identify the correlations all at once without using temporal information. Although being simple, they are unable to learn the intra-modal and cross-modal dynamics evolving along the time sequence, thus suffering from the accuracy loss for prediction.

**Multimodal temporal models**, on the other hand, handle multimodal interactions at a much finer granularity along the time dimension. The long-short term memory (LSTM) [10] has been extensively used for the sequential multimodal setting. Among them, multi-view LSTM (MV-LSTM) [24] partitions the memory cell corresponding to specific modality to capture both view-specific and cross-view interactions; bidirectional contextual LSTM (BC-LSTM) [23] is proposed to conduct context-dependent sentiment analysis and emotion recognition with multimodal time series. The memory fusion network (MFN) [28] stores cross-modal and intra-modal interactions along time domain with a multi-view gated memory, while the multi-attention recurrent network (MARN) [29] employs a multi-attention block to discover cross-modality dynamics with attention coefficients. More recently, the recurrent multistage fusion network (RMFN) [14] decomposes the fusion into multiple stages, with each focusing on a subset of signals whose fusion outcomes build upon intermediate representations of previous stages. However, compared with tensor based multimodal fusions, all above approaches are limited to model only the linear interactions, unable to identify complicated multimodal correlations.

# 5 Experiments

## 5.1 Experiment setups

**Datasets.** CMU-MOSI dataset [30] consists of $2,199$ opinion video clips from YouTube movie reviews. Each clip is assigned with a specified sentiment in the range $[-3, 3]$ from high negative to high positive. There are $1,284$ segments in the train set, 229 segments in the validation set and 686 segments in the test set. IEMOCAP dataset [2] contains a total number of 302 videos. The segments from videos are annotated with discrete emotions (neutral, fear, happy, angry, disappointed, sad, frustrated, excited, surprised), as well as dominance, valance and arousal. The division of the train, validation and test sets is $6,373$, $1,775$ and $1,807$, respectively. The splits of two datasets are speaker-independent, ensuring the specified speaker can only belong to one of the three sets.

**Features.** For IEMOCAP, we adopt the preprocessed non-temporal inputs following the work of LMF [17], in which the acoustic and visual features are obtained by averaging out the time dimension. For CMU-MOSI: temporal features are utilized in the same way as MFN [28], where the extracted features of three modalities are synchronized at word level in accordance with the text modality.

**Comparisons.** We include the following cutting-edge tensor and non-tensor based models into our comparisons with HPFN: memory fusion network (MFN) [28], multi-attention recurrent network (MARN) [29], tensor fusion network (TFN) [27] and low-rank multimodal fusion network (LMF) [17], as well as some other baselines. We report mean absolute error (MAE) and Pearson correlation, accuracy and F1 measure. For our HPFN, the evaluations are repeated 5 times for the optimal settings.

**Model architectures.** The architectures of HPFN adopted in our experiments are described in Table 2, including the two-layer densely connected variants HPFN-L2-S1 and HPFN-L2-S2 .

Table 3: Results for sentiment analysis on CMU-MOSI and emotion recognition on IEMOCAP.

| Models | CMU-MOSI | | | | | IEMOCAP | | | |
|---|---|---|---|---|---|---|---|---|---|
| | MAE | Corr | Acc-2 | F1 | Acc-7 | F1-Happy | F1-Sad | F1-Angry | F1-Neutral |
| SVM [6] | 1.864 | 0.057 | 50.2 | 50.1 | 17.5 | 81.5 | 78.8 | 82.4 | 64.9 |
| DF [20] | 1.143 | 0.518 | 72.3 | 72.1 | 26.8 | 81.0 | 81.2 | 65.4 | 44.0 |
| BC-LSTM [23] | 1.079 | 0.581 | 73.9 | 73.9 | 28.7 | 81.7 | 81.7 | 84.2 | 64.1 |
| MV-LSTM [24] | 1.019 | 0.601 | 73.9 | 74.0 | 33.2 | 81.3 | 74.0 | 84.3 | 66.7 |
| MARN [29] | 0.968 | 0.625 | 77.1 | 77.0 | 34.7 | 83.6 | 81.2 | 84.2 | 65.9 |
| MFN [28] | 0.965 | 0.632 | 77.4 | 77.3 | 34.1 | 84.0 | 82.1 | 83.7 | 69.2 |
| TFN [27] | 0.970 | 0.633 | 73.9 | 73.4 | 32.1 | 83.6 | 82.8 | 84.2 | 65.4 |
| LMF [17] | **0.912** | 0.668 | 76.4 | 75.7 | 32.8 | 85.8 | 85.9 | **89.0** | 71.7 |
| HPFN, P=[4] (audio) | 1.404 | 0.223 | 57.3 | 57.4 | 19.0 | 79.4 | 81.8 | 84.9 | 63.6 |
| HPFN, P=[4] (video) | 1.409 | 0.221 | 57.0 | 57.1 | 20.6 | 83.2 | 73.2 | 72.3 | 58.5 |
| HPFN, P=[4] (text) | 0.975 | 0.634 | 76.4 | 76.4 | 35.1 | 85.3 | 83.0 | 85.6 | 70.8 |
| HPFN, P=[4] | 0.965 | 0.650 | **77.5** | **77.4** | 36.0 | 85.7 | 86.4 | 88.3 | 72.1 |
| HPFN, P=[8] | 0.968 | 0.648 | 77.2 | 77.2 | **36.9** | 85.7 | 86.5 | 87.9 | 71.8 |
| HPFN-L2, P=[2, 2] | 0.945 | **0.672** | **77.5** | **77.4** | 36.7 | **86.2** | **86.6** | 88.8 | **72.5** |

**Implementation details.** Following LMF [17], we use CP format as the 'workhorse' low-rank TN in our experiments for weight compression in PTP. The candidate CP ranks are $\{1, 4, 8, 16\}$. Other TNs variants will be investigated in future work. Since HPFN involves high-order moments when calculating element-wise multiplication, the values of intermediate features may vary drastically and hence lead to unstable predictions. To make the model numerically more stable, similar to [8], we could optionally apply power normalization (element-wise signed squared root) or $l_2$ normalization.

## 5.2 Experimental results

**Performance comparison with state-of-the-art models.** We first compare with the baselines and the cutting-edge models on the tasks of sentiment analysis and emotion recognition. The bottom rows in Table 3 record the performance of our model. We see that ours (on multimodal data) outperform the competitors in most of the metrics. Particularly, on the sentiment task, our HPFN at 8th order exceeds the previous best MARN on the 'Acc-7' by a margin of $2.2\%$. The overall best results are achieved by HPFN-L2, which implies the superior expressive power and efficacy of the hierarchical fusion structures. It is also interesting to notice that, even fed with unimodal input (text), our HPFN of order 4 obtains much better 'Acc-7' (35.1) and 'F1-Neutral' (70.8) precisions than almost all other methods, indicating the benefits brought by modelling high-order interactions.

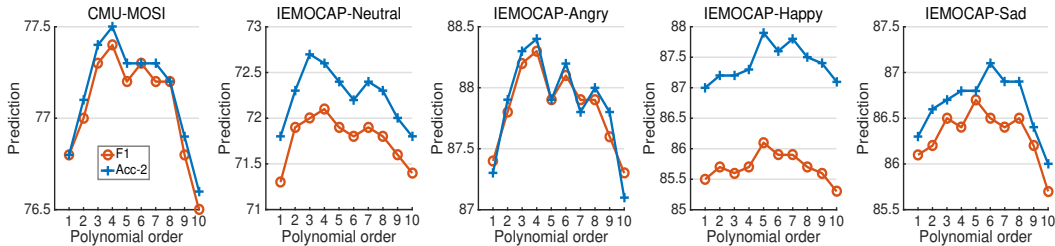

Figure 5: Results of the effect of orders of polynomial interactions on IEMOCAP and CMU-MOSI.

**Effect of the order of polynomial fusion.** As high-order moments play a critical role in our fusion strategy, we are interested to examine how distinct orders affect the predictive performance. For simplicity, we directly apply HPFN with power normalization to the non-temporal multimodal features (via averaging out the time dimension). The order $P$ varies from 1 to 10. In Figure 5, HPFN is able to achieve fairly good accuracies w.r.t. the tested orders. In particular, we can see HPFN maximizes predictions at the order 4 for the case of CMU-MOSI. For IEMOCAP, we observe the relatively higher performance peak at the orders of 3 and 4 in the 'neutral' and 'angry' emotions. As for the rest emotions, the desirable orders range 5 from 8. These observations signify the necessity and effectiveness of exploring high-order interactions in fusing multimodal features.

**Effect of the depth and dense connectivity.** In this part, we investigate the impact of various architecture designs, i.e., depth and dense connectivity, on the predictive performance. To focus on the change of the depth, we apply architectures to non-temporal multimodal features. For the depth

Table 4: Results of HPFN on non-temporal multimodal features w.r.t. the depth and dense connectivity.

| Models | IEMOCAP | | | | CMU-MOSI | | | | |
|---|---|---|---|---|---|---|---|---|---|
| | F1-Happy | F1-Sad | F1-Angry | F1-Neutral | MAE | Corr | Acc-2 | F1 | Acc-7 |
| HPFN, P=[2] | 85.7 | 86.2 | 87.8 | 71.9 | 0.973 | 0.635 | 77.1 | 77.0 | 35.9 |
| HPFN-L2, P=[2, 2] | 86.2 | 86.6 | 88.8 | 72.5 | 0.958 | 0.652 | 77.1 | 77.1 | 36.3 |
| HPFN-L2-S1, P=[2, 2] | 86.2 | 86.7 | 88.9 | 72.6 | 0.959 | 0.654 | **77.3** | 77.2 | **36.5** |
| HPFN-L2-S2, P=[2, 2] | **86.2** | 86.7 | **89.0** | **72.7** | **0.957** | **0.656** | 77.3 | **77.3** | 36.5 |
| HPFN-L3, P=[2, 2, 1] | 86.1 | **86.8** | 88.3 | 72.7 | 0.960 | 0.651 | 76.8 | 76.8 | 36.0 |
| HPFN-L4, P=[2, 2, 2, 1] | 85.8 | 86.4 | 88.1 | 72.5 | 0.992 | 0.634 | 76.6 | 76.5 | 34.6 |

Table 5: Results on the modelling of locally mixed temporal-modality features.

| Models | CMU-MOSI | | | | |
|---|---|---|---|---|---|
| | MAE | Corr | Acc-2 | F1 | Acc-7 |
| HPFN-L2, P=[2, 2] (non-temporal) | 0.958 | 0.652 | 77.1 | 77.1 | 36.3 |
| HPFN-L2, P=[2, 2] (temporal-overlapped, audio) | 1.407 | 0.229 | 57.4 | 56.2 | 20.1 |
| HPFN-L2, P=[2, 2] (temporal-overlapped, video) | 1.358 | 0.183 | 61.2 | 61.3 | 20.3 |
| HPFN-L2, P=[2, 2] (temporal-overlapped, text) | **0.933** | 0.677 | 76.7 | 76.6 | 35.4 |
| HPFN-L2, P=[2, 2] (temporal-overlapped) | 0.944 | **0.678** | **77.5** | **77.4** | **36.7** |
| HPFN-L2, P=[2, 2] (weight-shared) | 0.955 | 0.667 | 77.0 | 76.9 | 35.7 |

variants, we validate on `HPFN`, `HPFN-L2`, `HPFN-L3` and `HPFN-L4`. We also compare with two densely connected variants: `HPFN-L2-S1` and `HPFN-L2-S2`. In Table 4, we find two-layer and three-layer based architectures attain the better overall results than their both one-layer and four-layer counterparts. In particular, `HPFN-L2-S2` reach the best precisions on both datasets. The `HPFN` is too simple to learn the complex interactions while `HPFN-L4` containing too many intermediate nodes is likely to overfit for this specific architecture design. Allowing skip connections further enhances the performance of `HPFN-L2`, which may be due to the incorporation of the guidance from the more discriminative unimodal signals without adding more intermediate layers.

**Effect of the modelling mixed temporal-modality features.** Being able to deal with a local mixture of temporal-modality features is one desirable property of our model. In this test, we examine how the model behaves by considering both temporal and modality domains. We adapt `HPFN-L2` to the temporal context with 'window' size of $[4 \times 2]$ for the input layer, and set the stride step as $2$ along the time dimension. The non-temporal `HPFN-L2` only considers modality domain, by averaging out the time dimension. Table 5 indicates the superiority of the temporal `HPFN-L2` over the non-temporal one. We further attempt to share the PTPs by scanning the 'window' along the temporal direction. It turns out that sharing a single PTP unit for multiple windows does not bring the extra performance gain for this setting. Figure 6 displays the predictions w.r.t. the 'window' size in the temporal domain. For non-weight-shared case, moderate 'windows' (sizes of 5 and 10) reach the peak performance. In contrast, weight-shared modal gets the relatively high performance under the largest window size (20). This again implies sharing with a single PTP may not be able to capture the local, evolving dynamics of interactions.

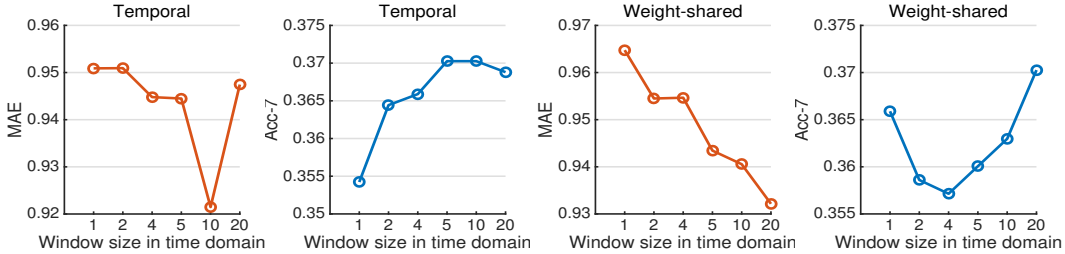

Figure 6: Results on predictions w.r.t. the 'window' size along the time domain. The left two figures: no-weight-shared model. The right two figures: weight-shared model.

## 6  Conclusion

In this paper, we proposed a high-order polynomial multilinear pooling block for multimodal feature fusion. Based on this, we established a hierarchical polynomial fusion network (HPFN) which can

flexibly fuse the mixed features across both time and modality domains. The proposed model is effective in capturing much complex temporal-modality correlations from local scale to global scale. The various experiments on real multimodal fusion tasks validate the superior performance of the proposed model. For future work, we like to further examine how the architecture designs affect the prediction performance. For example, attaching multiple PTP blocks to a single 'window', and sharing those multiple PTP 'fusion filters' along the time dimension to model more complex patterns of correlations.

## Acknowledgments

This work was partially supported by JSPS KAKENHI (Grant No. 17K00326), the national key research and development program intergovernmental international science and technology innovation cooperation project (MOST-RIKEN) under Grant 2017YFE0116800 and the national natural science foundation of China (Grant No. 61633010).

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
