[Supplementary Material · neurips_2019_sup.pdf]

# Deep Multimodal Multilinear Fusion with High-order Polynomial Pooling Supplemental Materials

## 1 Experimental Setups

**Features.** We follow LMF [1] and MFN [4] to adopt the preprocessed multimodal features as the input of our model. Specifically, we use pre-trained Glove world embeddings [2] to represent the spoken texts. The visual features such as facial action units, facial landmarks and head pose are extracted by the Facet library. The acoustic features obtained using COVAREP includes mel frequency cepstral coefficients, pitch, glottal source, peak slope and so on. The alignment is carried out by P2FA [3] to align the features of three modalities at the word granularity. As a result, for the MOSI dataset, each utterance is divided into 20 time steps. At each step, we obtain the temporal spoken language, visual and acoustic features with the respective lengths of 300, 20 and 5. As for the IEMOCAP dataset, the dimensionality of non-temporal visual and acoustic features 35 and 74.

**More implementation details.** For simplicity, we adopt the CP format to compress the tensor weight in PTP block. The CP rank is the hyper-parameter and chosen from the set $\{1, 4, 8, 16\}$. Other TNs formats such as TT and TR can also be used for the compression. For the temporal case, we also apply one-layer LSTM on the input features before feeding them to the model. We set the time window at each layer as a hyper-parameter. For example, in the case of non-overlapping time window, if the time steps of utterance is 20, then the possible choices could be $\{1, 2, 4, 5, 10, 20\}$. The dimensionality of the fused feature at each layer is also a hyper-parameter that is selected from the candidates $\{30, 40, 50, 60, 70\}$. For rest tuning parameters, the learning rate is chosen from the set $\{0.0003, 0.0005, 0.001, 0.003\}$ with the weight decay rate from set $\{0, 0.001, 0.002, 0.01\}$.