[Reviews · NeurIPS 2019]

Reviewer 1



Originality: The paper proposes a novel enhancement of tensor based multimodal fusion that enables accounting for local effects and a hierarchical architecture that folds in the proposed enhancement. Quality: The proposed technique is technically sound. Thorough literature survey and positioning of the paper with respect to the literature. Clarity: Very systematically and clearly written paper. Very easy to read. Significance: Strong impact. Clear advancement of the state of the art.

Reviewer 2



The preliminaries section lays out the required mathematical formulations of tensor networks. While the section presented serves its function, it could potentially be more clear if the authors spaced out and typeset the maths (similar to how it is done in Section 3.1). The visualizations in Figure 2/3 illustrates how a fusion network (and hierarchical network) can be constructed with PTP units. These visualizations clearly communicate how features are pooled across modality and time step. That said, perhaps the descriptions about the HPFN (Section 3.2) are overly verbose. Analogies (such as the analogy connecting ideas to CNN architectures, L128-131) are helpful, as are the details comparing and contrasting connections to ConvACs [5] and DenseNets [11]. The related work section further details comparable architectures and limitations of previous work. Specifically, the authors note that the proposed architecture should be 'more' capable of capturing complicated multimodal correlations than comparable baselines. The experiments presented in the paper cover the CMU-MOSI dataset [28] and the IEMOCAP dataset [2]. Both of these experiments are sentiment classification from text, acoustic, and/or visual features. For baseline comparison, the authors adopt similar featurization methodologies to previous work. The model architectures section (L210-225) is not clear. I would like the authors to consider a table which would allow for reference of each architecture and respective details. The results are clearly presented in Table 1. I would like some measure of significance testing reported in the paper. The results are compelling and support the hypothesis that modelling high-order interactions is beneficial toward these classification tasks. Details on how the models were trained, and the parameters were optimized is notably missing from the paper. Additionally, the empirical computational overhead is unclear. While the authors experimented with a collection of models, they do not compare the number of parameters, training time, or training curves of each of the models. This is likely important results given that the authors are arguing the performance benefits of the PTP units.

Reviewer 3



****************************Originality**************************** Strengths: - This paper opens up the new direction of considering higher-order interactions between multiple modalities. It can be seen as a novel application of existing techniques in tensors to multimodal learning. Weaknesses: - The paper can be seen as incremental improvements on previous work that has used simple tensor products to representation multimodal data. This paper largely follows previous setups but instead proposes to use higher-order tensor products. ****************************Quality**************************** Strengths: - The paper performs good empirical analysis. They have been thorough in comparing with some of the existing state-of-the-art models for multimodal fusion including those from 2018 and 2019. Their model shows consistent improvements across 2 multimodal datasets. - The authors provide a nice study of the effect of polynomial tensor order on prediction performance and show that accuracy increases up to a point. Weaknesses: - There are a few baselines that could also be worth comparing to such as “Strong and Simple Baselines for Multimodal Utterance Embeddings, NAACL 2019” - Since the model has connections to convolutional arithmetic units then ConvACs can also be a baseline for comparison. Given that you mention that “resulting in a correspondence of our HPFN to an even deeper ConAC”, it would be interesting to see a comparison table of depth with respect to performance. What depth is needed to learning “flexible and higher-order local and global intercorrelations“? - With respect to Figure 5, why do you think accuracy starts to drop after a certain order of around 4-5? Is it due to overfitting? - Do you think it is possible to dynamically determine the optimal order for fusion? It seems that the order corresponding to the best performance is different for different datasets and metrics, without a clear pattern or explanation. - The model does seem to perform well but there seem to be much more parameters in the model especially as the model consists of more layers. Could you comment on these tradeoffs including time and space complexity? - What are the impacts on the model when multimodal data is imperfect, such as when certain modalities are missing? Since the model builds higher-order interactions, does missing data at the input level lead to compounding effects that further affect the polynomial tensors being constructed, or is the model able to leverage additional modalities to help infer the missing ones? - How can the model be modified to remain useful when there are noisy or missing modalities? - Some more qualitative evaluation would be nice. Where does the improvement in performance come from? What exactly does the model pick up on? Are informative features compounded and highlighted across modalities? Are features being emphasized within a modality (i.e. better unimodal representations), or are better features being learned across modalities? ****************************Clarity**************************** Strengths: - The paper is well written with very informative Figures, especially Figures 1 and 2. - The paper gives a good introduction to tensors for those who are unfamiliar with the literature. Weaknesses: - The concept of local interactions is not as clear as the rest of the paper. Is it local in that it refers to the interactions within a time window, or is it local in that it is within the same modality? - It is unclear whether the improved results in Table 1 with respect to existing methods is due to higher-order interactions or due to more parameters. A column indicating the number of parameters for each model would be useful. - More experimental details such as neural networks and hyperparameters used should be included in the appendix. - Results should be averaged over multiple runs to determine statistical significance. - There are a few typos and stylistic issues: 1. line 2: "Despite of being compact” -> “Despite being compact” 2. line 56: “We refer multiway arrays” -> “We refer to multiway arrays” 3. line 158: “HPFN to a even deeper ConAC” -> “HPFN to an even deeper ConAC” 4. line 265: "Effect of the modelling mixed temporal-modality features." -> I'm not sure what this means, it's not grammatically correct. 5. equations (4) and (5) should use \left( and \right) for parenthesis. 6. and so on… ****************************Significance**************************** Strengths: - This paper will likely be a nice addition to the current models we have for processing multimodal data, especially since the results are quite promising. Weaknesses: - Not really a weakness, but there is a paper at ACL 2019 on "Learning Representations from Imperfect Time Series Data via Tensor Rank Regularization” which uses low-rank tensor representations as a method to regularize against noisy or imperfect multimodal time-series data. Could your method be combined with their regularization methods to ensure more robust multimodal predictions in the presence of noisy or imperfect multimodal data? - The paper in its current form presents a specific model for learning multimodal representations. To make it more significant, the polynomial pooling layer could be added to existing models and experiments showing consistent improvement over different model architectures. To be more concrete, the yellow, red, and green multimodal data in Figure 2a) can be raw time-series inputs, or they can be the outputs of recurrent units, transformer units, etc. Demonstrating that this layer can improve performance on top of different layers would be this work more significant for the research community. ****************************Post Rebuttal**************************** I appreciate the effort the authors have put into the rebuttal. Since I already liked the paper and the results are quite good, I am maintaining my score. I am not willing to give a higher score since the tasks are rather straightforward with well-studied baselines and tensor methods have already been used to some extent in multimodal learning, so this method is an improvement on top of existing ones.

[Author Response · NeurIPS 2019]

We thank all Reviewers for their constructive comments and insightful suggestions.

**To Reviewer 1:**

(1) Thank you for the very positive comments on our paper and kind suggestions about
TRECVID dataset, we believe more impressive results can be provided in final version.

**To Reviewer 2 & Reviewer 3:**

(1) Parameter complexity: for PTP, due to the symmetry of feature tensor as well as the
symmetric weight tensor, the number of parameters is independent of order $P$ and linearly scales with the concatenated
mixed features. For $L$-layer HPFN, number of parameters is linearly related to the number of 'windows' $\sum_{l=1}^{L} N_l$. In
practice, $N_l$ is usually small and decreasing along layers, e.g. $N_1 > N_2 > ... > N_L$. In our tests, complete/partial
sharing strategy makes $N_l$ even smaller. In principle, the parameter of HPFN is larger than or comparable to LMF (as
HPFN is more powerful with temporal modelling), but significantly less than TFN. Please refer to the Table in this page.

The tradeoff is, if we employ more layers (or with more intermediate nodes in each layer) we get greater expressive
capability. In practice, we need to choose optimal one so as not to overfit.

(2) Time complexity: PTP is comparable or similar to LMF; HPFN is less than $\sum_{l=1}^{L} N_l$ times of LMF, depending on
specific architecture design choices such as number of layers, number of windows, window size and etc.

(3) About training details, such as hyper-parameters settings and number of parameters, will be added in final version.

**To Reviewer 2:**

(1) Thanks for detailed comments on clarity of paper (such as maths layout, concise descriptions of HPFN, model
architecture table), we will include them in the final version.

(2) Regarding training curves, we illustrate HPFN L1 & L2 and LMF in Figure in this page showing that HPFN is better
than LMF. More comparisons on training curves will be added in the final version.

(3) It is a great suggestion to include significance testing, such as p-value test, in the paper.

**To Reviewer 3:**

(1) Regarding the concept of 'local interactions', it refers to interactions of the concatenated features, from a time
window and a subset of modalities located within a 'local window' (analogous to the 'convolution filter' of CNN).
Please see 'bounding rectangles' in Figure 3 in paper as an example. We will improve the clarity by rewriting this part.

(2) The originality of our work includes: (a) higher order interactions and (b) local temporal information fusion can
be achieved by PTP. HPFN can be explained as a CNN-style hierarchical fusion framework where the convolution
operation is replaced by the PTP operator, to identify cross-modal interactions through time sequence.

(3) Regarding more baselines, MMB1&2 [NAACL19'] are two wonderful baselines that learn embeddings of multimodal
utterances, we will add them in the next version. ConAC is based on convolutional operator and suitable for the
conventional image recognition task. It does not directly consider the deep fusion of multimodal data, hence might not
perform well. But it is interesting to adapt/modify ConAC to multimodal setting in the future work.

(4) Regarding imperfect data, our current model does not take imperfect data into account, but it is very interesting
direction to work on. A very recent T2FN [ACL19'] is an excellent approach for imperfect data. We will refer this work
and believe that incorporating its novel idea of tensor regularizer can make our method more robust to noisy data. The
sharing strategy among multiple 'fusion filters' at each layer might be another possible option.

(5) Regarding the depth, we test HPFN up to 4 layers (Table 2 in paper) and the optimal number of layers for IEMOCAP
and MOSI are 2 and 3, using the listed architectures. The optimal depth also depends on how each layer is designed.

(6) Regarding the order $P$, automatically determining the optimal $P$ is nontrivial and interesting to investigate. Now
the best way is to use CV. In our empirical test, the preferred order normally ranges from 4 to 8, which are related to
specific tasks and datasets.

(7) Regarding performance improvement, as shown in Table 1 in paper, the high-order pooling and the number of layers
(meaning more parameters) achieved similar amount of performance improvement over SOTA LMF method. We agree
with Reviewer that it would be nice by adding more qualitative analysis in the final version.

| Model | TFN [non-temporal] | LMF [non-temporal] | PTP [temporal] | HPFN (L layers) [temporal] |
|---|---|---|---|---|
| Parameter Complexity | $\mathcal{O}(I_y \prod_{m=1}^{M} I_m)$ | $\mathcal{O}(I_y r(\sum_{m=1}^{M} I_m))$ | $\mathcal{O}(I_y r(\sum_{t=1}^{T} \sum_{m=1}^{S} I_{t,m}))$ | $\mathcal{O}(I_y r(\sum_{l=1}^{L} N_l)(\sum_{t=1}^{T} \sum_{m=1}^{S} I_{t,m}))$ |

Table 1: $I_y$ is output length. $M$ is number of modalities. $r$ is tensor rank. For PTP, $[T, S]$ is the 'local window size' with $S \leq M$. $I_{t,m}$ is the dimension of features from modality $m$ at time $t$. For HPFN, $N_l$ is the number of PTP 'windows' at layer $l \in [L]$.

[Meta-Review · NeurIPS 2019]

The authors propose the high-order Polynomial Tensor Pooling (PTP) block, which can fuse locally mixed temporal features. A concrete neural architecture utilizing the PTP block as the main component shows good performance. To avoid the exponential parameter explosion of tensor product representations, the authors present a novel technique that does not restrict the order of feature interaction. Although there are many details that can be improved, including a more rigorous evaluation, the significance and originality of the work makes this a promising paper for NeurIPS.